# Constitutive Model of the Surface Roughening Behavior of Austenitic Stainless Steel

**DOI:** 10.3390/ma15124348

**Published:** 2022-06-20

**Authors:** Abdul Aziz, Ming Yang, Tetsuhide Shimizu, Tsuyoshi Furushima

**Affiliations:** 1Graduate School of System Design, Tokyo Metropolitan University, 6-6-Asahigaoka, Hino, Tokyo 191-0055, Japan; yang@tmu.ac.jp (M.Y.); simizu-tetuhide@tmu.ac.jp (T.S.); 2Department of Mechanical and Biofunctional Systems, Institute of Industrial Science, University of Tokyo, 4 Chome-6-1 Komaba, Meguro, Tokyo 153-8505, Japan; tsuyoful@iis.u-tokyo.ac.jp

**Keywords:** surface roughening, martensitic phase transformation (MPT), work hardening (*n*)

## Abstract

The martensitic phase transformation (MPT) is one of the most important factors that enhances the surface roughening of stainless-steel thin metal foils (TMF), such as SUS 304, compared to others without MPT, even in the same plastic strain. However, the conventional roughening model does not take into account the influence of MPT. In this study, the authors proposed a new constitutive model to express the surface roughening by taking the influence of MPT into account. The volume fractions of MPT for TMF of SUS304 in various grain sizes are accounted for quantitatively after the tensile test at room temperature and an elevated temperature, and the effect of MPT on the surface roughening is evaluated in comparison to using TMF of SUS316, in which MPT does not occur during plastic deformation. Then, a constitutive model of the surface roughening based on the experimental results is successfully built.

## 1. Introduction

Micro-/meso-scaled parts have been widely used in electronics, automobiles, aerospace, and biomedicine in tandem with the ubiquitous trend of product miniaturization [1,2,3,4,5,6]. In this decade, the demand for micro parts has increased unabated. Stainless steels such as SUS 304 and SUS 316 have wide applications in the micro-manufacturing industry. The estimated rise in turnover from USD 15 to 35 billion in the last seven years [1,2,3,4,5,6,7] shows a growing demand for micro technical products, which is mainly driven by the rising trend of the miniaturization of products. The demand for miniaturization comes not only from customers, who desire more handy electronic devices and more integrated functions, but also from technical applications such as medical equipment, sensor technology, and optoelectronics [3,4,8]. All these products contain mechanical parts such as leverages, connector pins, resistor caps, screws, contact springs, and cheap lead frames [1,2,3,4,9,10].

To understand the mechanisms behind surface roughening in austenitic stainless steel, many studies have been conducted on stainless steel sheets [2,4,11,12,13,14]. It has been generally reported that surface roughening linearly increases with plastic strain [14,15,16,17] and is affected not only by plastic deformation conditions such as the strain ratio or strain path but also by material factors such as the grain size crystal structure, crystal orientation and texture [15,17,18,19]. When the austenitic stainless steel is subjected to plastic deformation, martensitic phase transformation (MPT) occurs [15,20,21,22]. MPT occurs as a result of the slip band intersection. The slip band intersection is the place of the martensitic embryo and then becomes MPT [23,24,25,26,27,28,29,30,31,32,33,34,35]. 

In early studies, surface roughening was examined experimentally. Yamaguchi et al. [17] showed that the degree of surface roughening in aluminum-killed steel sheets depended on the amount of equivalent plastic strain and the crystal grain size. Mahmudi et al. [18] concluded that surface roughening was caused by inhomogeneity of the material’s microstructure. The degree of development in surface roughening also depended on the strain rate and the crystal structure [5,6,7,8,9,10,11,12,13,14,15]. The rotation of the crystal orientation caused by plastic strain was the factor of surface roughening in steel and aluminum sheets. [6,7,8,9,10,11,12,13,14,15]. The deformation of interior crystal grains from the surface to the second layer could affect the surface unevenness in some kinds of metal [6,7,8,9,10,11,12,13,14,15]. The previous studies concluded that the surface roughening should be expressed as a function of plastic strain as ΔRa = Ce_p_, where ΔRa is the increase in surface roughness, C is a constant, and e_p_ is the plastic strain level [25,26,27,28,29,30,31,32,33,34].

Furushima et al. [34] also concluded that surface roughening not only depends on the strain level but also depends on grain size. The surface roughening should be expressed as follows: ΔRa = Ce_p_ · Dg/t(1)
where ΔRa is the increase in surface roughness, e_p_ is the plastic strain level, Dg is the grain size, and t is the thickness of the workpiece.

The authors have been working on SUS 304 and SUS 316 thin metal foils and found that surface roughening not only depends on the strain level and grain size but also depends on martensitic phase transformation (MPT) and grain misorientation (GMO). Thus, it is not enough if the surface roughening depends on the grain size and strain level. MPT is the one factor in SUS 304 thin metal foil (TMF) that enhances the surface roughening compared to SUS 316 TMF, in which the surface roughening in SUS 304 increases higher than SUS 316 TMF with the same grain size (Dg) and the same strain level at room temperature [20]. Furthermore, the influential factor of grain size does not explain the surface roughening behavior in thin metal foils since the influential factors such as MPT and GMO are also related to the grain size of the material. There is currently a lack of studies that create a theoretical model to express the effect of MPT and GMO on the surface roughening in austenitic stainless-steel foils. No theoretical model has determined the effect of MPT and GMO on the surface roughening behavior in various grain sizes. Thus, this study aims to build a theoretical model to express the effect of MPT and GMO on the surface roughening behavior with various grain sizes based on the experimental results of tensile tests using SUS 304 and SUS 316 thin metal foils (TMF).

## 2. Materials and Methods

### 2.1. Material

The result of the tensile test in our previous studies was utilized to analyze the volume fraction of MPT and GMO in this study. In this research, the author uses SUS 304 and SUS 316 thin metal foils of SUS 304 and SUS 316. The material condition for room [20] and elevated temperature [21] was similar to previous results, such as sample dimension, microstructure, grain size, and chemical composition. Room temperature was used to investigate the effect of MPT and GMO on the surface roughening behavior. High temperatures were used to investigate effect of resistance heating (RH) on the surface roughening behavior. Author uses two kinds of materials, SUS 304 and SUS 316. The thickness was 0.1 mm, the grain size (Dg) was 1.5 μm; SUS 304 and SUS 316 thin metal foils were 3.0 μm and 9.0 μm, respectively.

### 2.2. Method

The methodology of using room temperature [20] and elevated temperatures [21] has been explained in previous research. The methodology, in general, involves using uniaxial tensile stress state for five steps, and surface roughness is measured every step. After surface roughness is measured for five steps, phase transformation and grain misorientation are observed using SEM-EBSD method.

The surface roughening behavior was investigated at elevated temperatures using resistance heating method. The resistance heating was conducted at 500 °C for three minutes for every step of the tensile test. The emissivity of the sample was 0.94. The thermos sensor camera used in this experiment was Optris Xi-400, made in Japan. The uniaxial tensile test used the AGX-50KNVD machine, with loading capacity of up to 50 KN. The uniaxial tensile test machine was produced by Shimadzu, Japan. The surface roughening behavior was investigated using OLS-5000 laser microscope produced by Olympus Co., Tokyo, Japan.

The volume fraction of MPT and GMO was calculated using the ImageJ technique, utilizing the volume fraction based on color. The technique for calculating dσ/dε slopes involved obtaining value of σ and value of ε.

## 3. Construction of the Constitutive Model

### 3.1. Influence of the Volume Fraction of MPT on Surface Roughening

The surface roughening increases with the strain level. Furthermore, the surface roughening in SUS 304 increases higher than in SUS 316 TMF coarse grain at room temperature [20]. The main reason for this is that the MPT spreads inhomogeneously in the case of coarse grains. Figure 1 shows that the MPT spreads homogeneously in fine grains, as shown in Figure 1a, and spreads inhomogeneously in coarse grains, as shown in Figure 1b [20].

The hardness structure with the martensitic phase is higher than that with grain misorientation at the same strain level, and the hardness of the material increases with the martensitic phase volume fraction [22]. The hardness increases hyperbolically with the FMPT. The increasing MPT affects the increasing FMPT and grain strength. The grain that contains MPT has a higher grain strength than the grain without MPT [22].

Figure 2 shows the volume fraction of martensitic phase transformation (F_MPT_) at room temperature, and Figure 3 shows that at elevated temperatures. In Figure 2, the volume fraction of MPT (F_MPT_) in SUS 304 TMF increases linearly with the strain level. The FMPT in the fine grains of SUS 304 increases higher than in coarse grains of SUS 304. MPT occurs due to the slip band intersection in the grains during the uniaxial tensile test. The probability of the slip band intersection in the fine grains is higher than in the coarse grains; thus, the F_MPT_ in fine grains becomes much higher than in coarse grains.

Figure 3 shows that the F_MPT_ only increases linearly in the fine grains of SUS 304 TMF. The F_MPT_ was reduced significantly compared to room temperature. The F_MPT_ disappeared in coarse grains of SUS 304 TMF; because of the heating, the mobility of slip band movement increased, and the probability of slip band intersection decreased [21,22,23,24,25,26,27]. There is no F_MPT_ in SUS 316 TMF during the uniaxial tensile test, which uses resistance heating (RH).

Figure 4 and Figure 5 show the KAM map in SUS 316 and SUS 304 coarse grains at room temperature [20]. In the KAM map, there are changes in FGMO until the 25.0% strain level.

Figure 6 and Figure 7 show the GMO fraction (FGMO) in room temperature and elevated temperature. the FGMO increase proportional both in room and at elevated temperature. The FGMO decrease with increasing temperature both in fine and coarse grain. the FGMO in fine grains are higher than coarse grain both in room temperature (Figure 6) and at elevated temperature (Figure 7).

The grain misorientation (GMO) is obtained from the calculation of the grain misorientation in the center of the grains and then averaged, which can be obtained from the KAM map of the SEM-EBSD analysis of the structures after deformation [20]. In this study, the volume fraction of GMO means the amount of GMO in a certain area. Figure 4 and Figure 5 show a typical KAM map for SUS304 and SU316 TMFs in coarse grains from the KAM map of SEM-EBSD analysis in a previous study [20].

The FGMO counted for SUS 304 and SUS 316 TMF at room temperature and RH conditions shown in Figure 4 and Figure 5. The F_GMO_ decreases with the grain size (Dg) and increases higher in SUS 316 compared to the SUS 304 TMF both in fine and coarse grains [20]. The GMO spread more inhomogeneously for the coarse grains of SUS 316 than the SUS 304 TMF. However, the surface roughness in SUS 304 coarse grains was higher than in SUS 316 TMF coarse grains since MPT only occurred at the same strain level in SUS 304 TMF. The higher surface roughness in SUS 304 could be affected by the MPT that spreads inhomogeneously in the grains at room temperature.

The quantity of GMOs decreases because of resistance heating both in SUS 304 and SUS 316 TMF, as shown in Figure 5. The F_GMO_ in the coarse grains of SUS 304 TMF decreases at high temperatures compared to room temperature. The F_GMO_ in SUS 316 TMF was higher than in SUS 304 TMF, and the inhomogeneous grain strength in SUS 316 TMF was higher than in SUS 304 at elevated temperatures. As a result, the surface roughness in SUS 316 TMF was higher than in SUS 304 TMF [21].

Dislocation density, F_MPT_, and F_GMO_ increase proportionally with the strain level [24,25,26,27,28,29]. For fine grains, the FMPT and dislocation density in a grain become much higher than for coarse grains because the slip band intersection probability becomes much higher in fine grains than in coarse grains [24,27].

On the other hand, the work hardening index *n* is a macroscale view that may relate to surface roughening behavior. It is well known that when the *n* value of a material is higher than other material, the strain distribution is more uniform in the material than in the other one. Figure 8 and Figure 9 show the work hardening index *n* for SUS 304 and SUS 316 TMF. The *n* value in SUS 304 at room temperature was lower than in SUS 316 TMF, but the hardness of SUS 304 TMF was higher than SUS 316 TMF at room temperature after the tensile test because the MPT occurred in SUS 304 TMF. In the elevated temperature (E.T.) condition, the *n* value of SUS 304 TMF was a little bit higher than SUS 316 TMF. It was found that there was no MPT in SUS 304 TMF, and the hardness of SUS 304 TMF and SUS 316 TMF became similar [21,22]. The work hardening for SUS 316 was higher than SUS 304 for every grain size (Dg) at room temperature. This means that SUS 316 TMF has a more uniform deformation distribution in the workpiece. When a uniform deformation distribution in the workpiece occurs, the deformation in the grain becomes more homogeneous. The homogeneous grain deformation lowers the increase in surface roughness compared to inhomogeneous grain deformation at the same strain level. As a result, at room temperature, SUS 304 TMF shows higher surface roughness than SUS 316 TMF with the same strain level. On the other hand, at elevated temperatures, the work hardening of SUS 304 was higher than SUS 316 TMF. This means the SUS 304 TMF grain deformation distribution in the workpiece becomes more uniform or more homogeneous [21]. The increase in surface roughness in SUS 316 may become higher than in SUS 304 TMF with the same strain level. The phenomena of work hardening at room and high temperatures were different according to the occurrence or not of MPT.

### 3.2. Quantitative Model

Based on the discussion in the previous section, the surface roughening for SUS304 is larger than that for SUS316 at the same strain level since both MPT and GMO occur in SUS304, but only GMO occurs in SUS316. The FMPT should be taken into account for the calculation of the surface roughening. However, the MPT particularly helps the coarse grain structure to become more inhomogeneous during plastic deformation. On the other hand, the GMO is also proportional to dislocation density but induces a uniform deformation. Therefore, the effect of MPT should be higher on the roughening mechanism than GMO in stainless steel, especially in SUS 304. When the MPT is spread uniformly, such as for fine grains [20], the surface roughening increase is very low compared to coarse grains with the same strain level. Therefore, MPT could be a factor that enhances the surface roughening behavior, and it is necessary to express surface roughening as a function of F_MPT_ besides F_GMO_. However, MPT was not considered in conventional models, and these models cannot sufficiently express the surface roughening behavior in thin metal foils of SUS 304. For confirmation, the typical conventional model shown by Equation (1) in the Introduction section was used to fix the experimental results in our previous studies [20,21]. Figure 10 shows the results for room temperature and Figure 11 for elevated temperature experiments. It is seen that there is a discrepancy in the slopes for SUS304 and SUS316 at room temperature, as shown in Figure 10, while the slopes show almost the same tendency in elevated temperatures, as shown in Figure 11. We can thus conclude that the conventional model can accurately fit the experimental results at elevated temperatures but is not accurate for the results of room-temperature experiments since MPT occurs in the latter.

Here, the authors attempt to build a new theoretical model that could express how the MPT and GMO affect the surface roughening behavior in SUS 304 and SUS 316 TMF with various grain sizes. Since the MPT enhances the surface roughness in SUS 304, F_MPT_ was utilized as an additional factor to improve the conventional theoretical model for the prediction of surface roughening. The new theoretical model for the prediction of surface roughening by considering MPT and GMO can be expressed as Equation (2).
ΔRa = (F_MPT_ + F_GMO_)·Dg/t(2)
F_mpt_ = C_mpt_ × (e_p_/(Dg)^1/2^)(3)
F_Gmo_ = C_Gmo_ × (e_p_/(Dg)^1/2^)(4)
where C_MPT_ and C_GMO_ are constants for the volume fraction of MPT and volume fraction of GMO, respectively.

By substituting Equations (3) and (4) into Equation (2), Equation (5) can be obtained.
ΔRa = (C_Mpt_ + C_Gmo_) × e_p_ × Dg^1/2^(5)

The new model is used to fit the experimental results in our previous studies again. The results are shown in Figure 12 and Figure 13. It can be seen that the fitting accuracy was improved for both SUS304 and SUS316 in various grains and conditions of the room and elevated temperatures.

The slope for SUS 304 TMF is higher than that of SUS316 at high strain levels at room temperature, but for low strain levels, the slopes are similar, as shown in Figure 12. In the condition of elevated temperature, the slope was similar at a low strain and a high strain level for all kinds of grain sizes, as shown in Figure 13. This means that the model is not yet accurate when a high-volume fraction of the MPT occurs and needs to be improved for this case.

The new model with ΔRa proportional to Dg^1/2^ is better than a conventional model with ΔRa proportional to Dg for the case of SUS 316 TMF without MPT. The different slope in the new model is lower than in the conventional model at room temperature. The slope of the new model of SUS 316 TMF coarse grains at room temperature is 0.056, and the slope of the previous model is 0.125 for coarse grains. The slope of SUS 304 TMF coarse grains in the new model is 0.08, and the slope of SUS 304 TMF coarse grains was 0.177 in the previous model. This means the new model is better than the previous model.

## 4. Conclusions

The volume fraction of MPT and GMO after the tensile test is accounted for, and both FMPT and FGMO are proportional to the plastic strain;The MPT is considered to be a factor that influenced the surface roughening since the SUS 304 TMF showed higher surface roughness than SUS 316 at room temperature, but SUS 304 and SUS 316 TMF were almost the same at elevated temperatures at the same strain level;A new theoretical model that takes FMPT and FGMO into account is proposed, and the difference in the calculated surface roughness for SUS304 and SUS316 at various grain sizes becomes smaller in the new theoretical model than in the conventional models.

## Figures and Tables

**Figure 1 materials-15-04348-f001:**
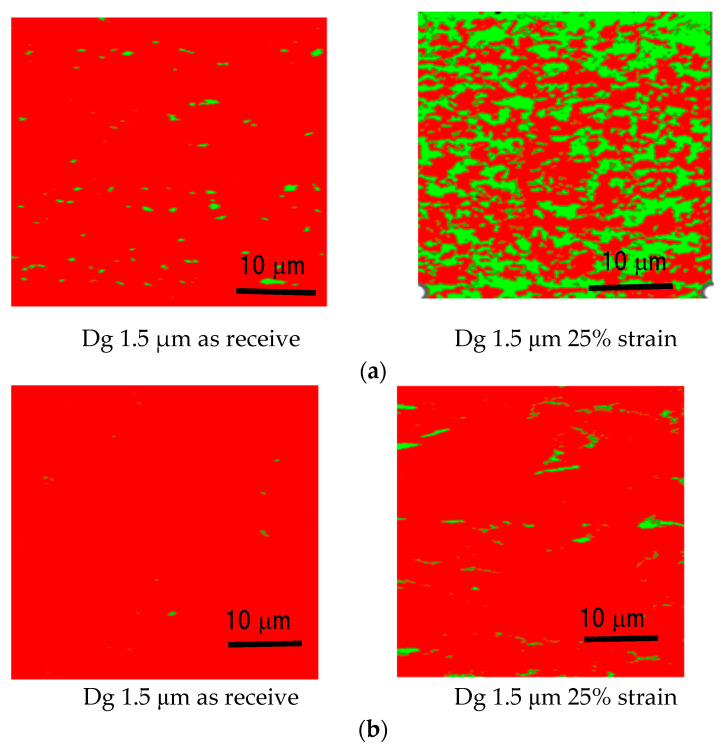
The phase map of SUS 304 fine grain and coarse grain [20]. (**a**) Fine grain; (**b**) Coarse grain.

**Figure 2 materials-15-04348-f002:**
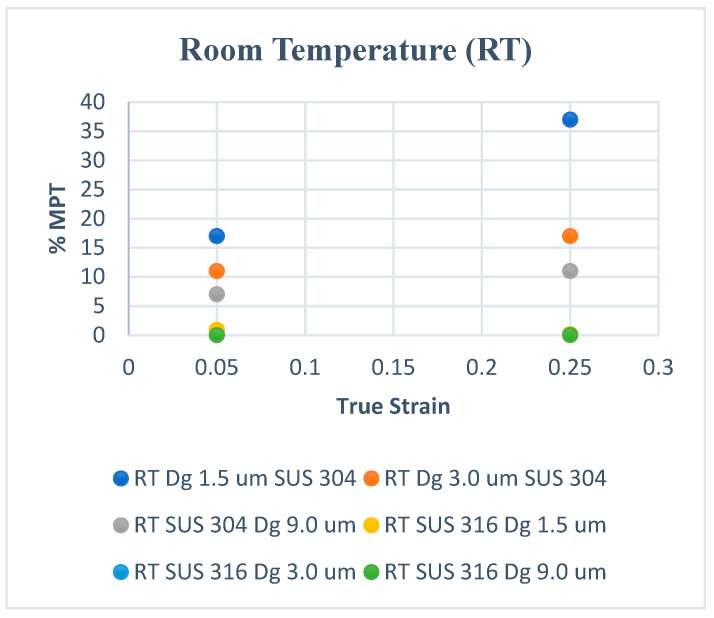
MPT at room temperature.

**Figure 3 materials-15-04348-f003:**
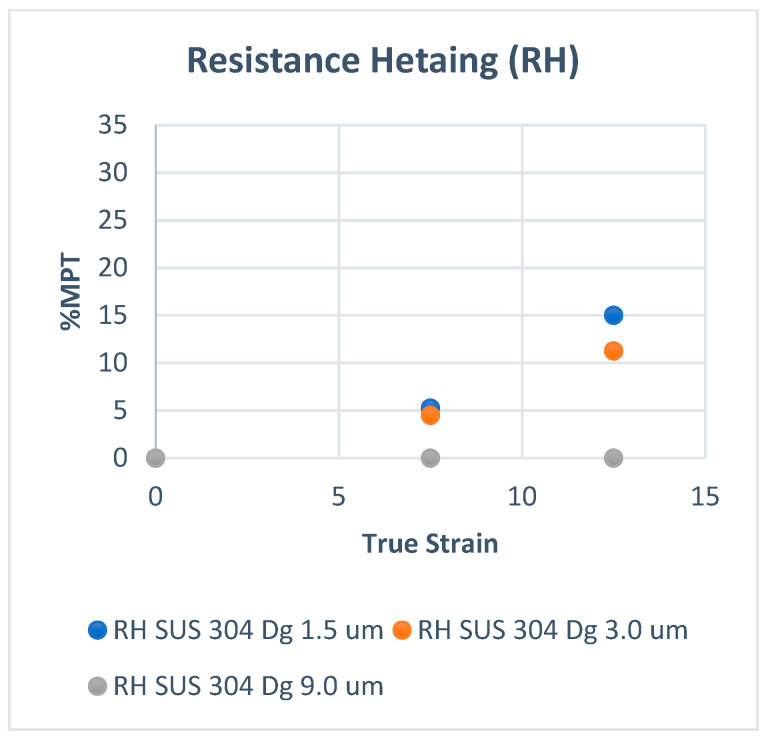
MPT at elevated temperatures.

**Figure 4 materials-15-04348-f004:**
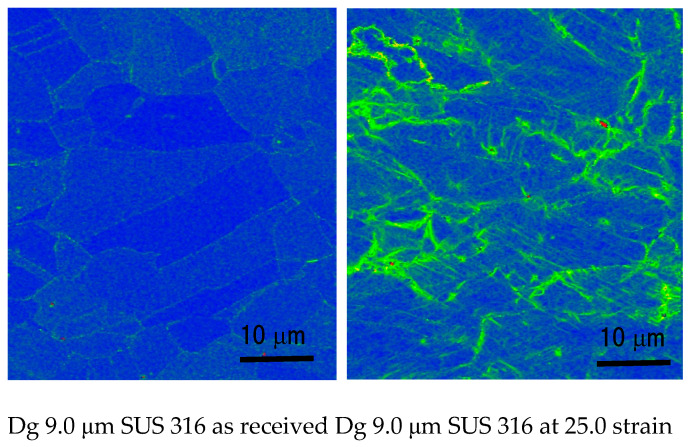
KAM map of SUS 316 TMF.

**Figure 5 materials-15-04348-f005:**
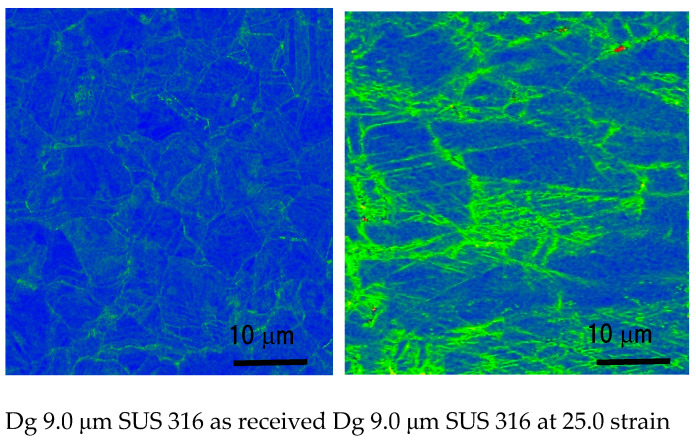
KAM map of SUS 304 TMF.

**Figure 6 materials-15-04348-f006:**
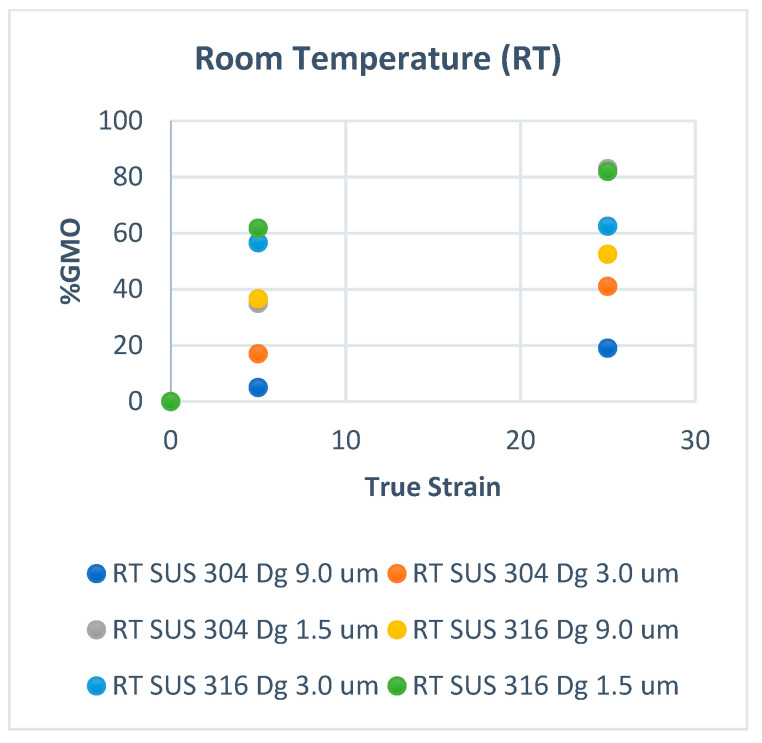
The FGMO at room temperature.

**Figure 7 materials-15-04348-f007:**
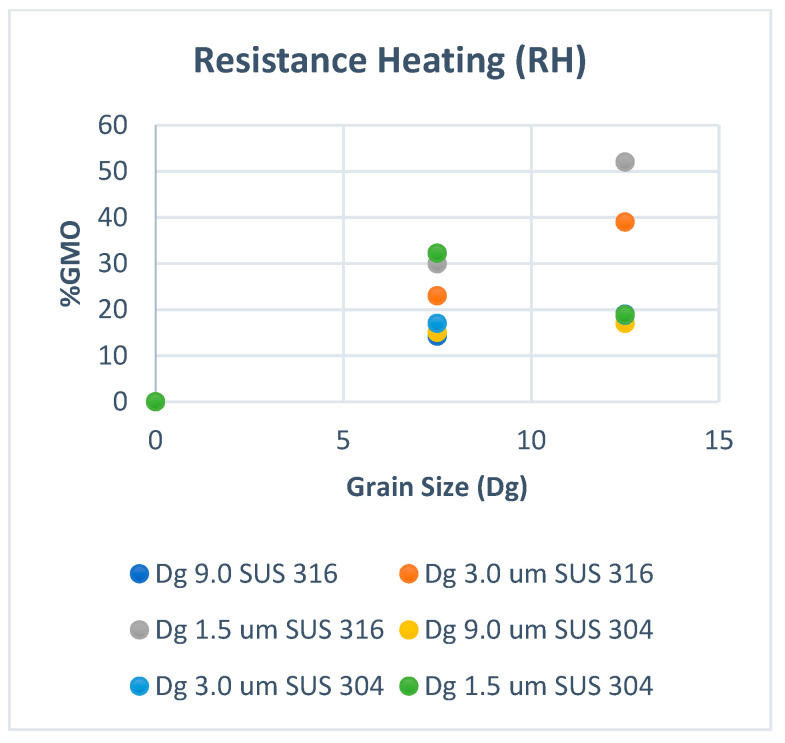
The FGMO with resistance heating (RH).

**Figure 8 materials-15-04348-f008:**
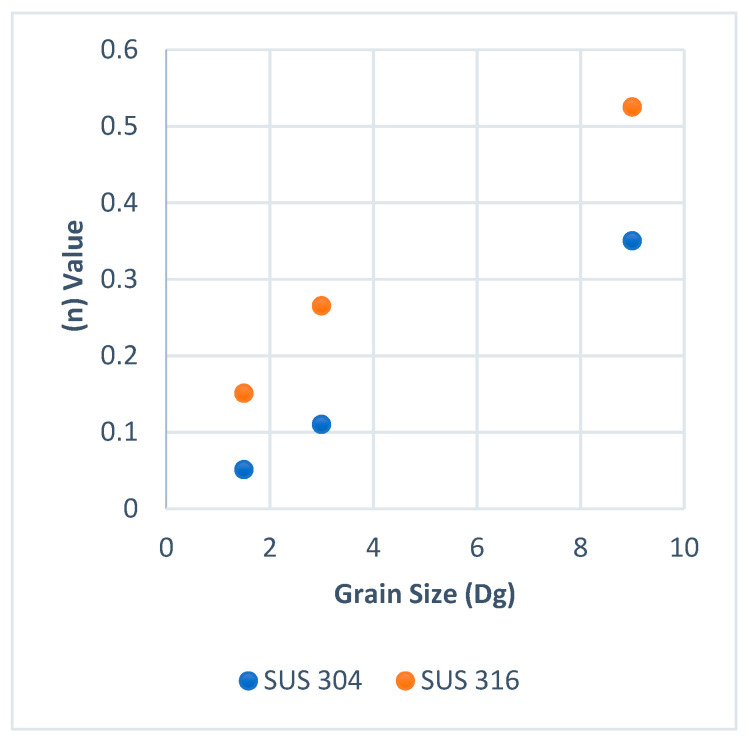
Work hardening room temperature.

**Figure 9 materials-15-04348-f009:**
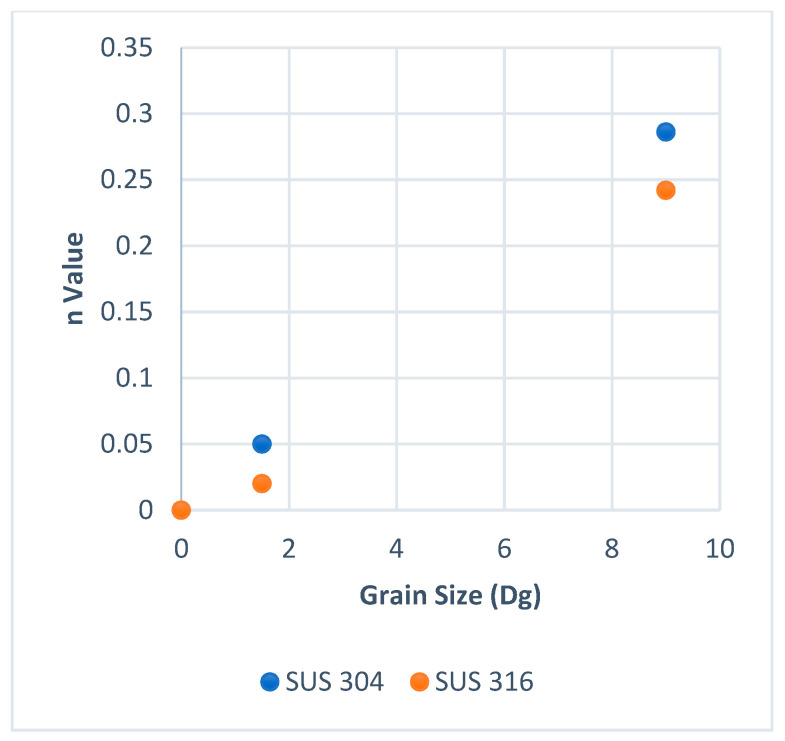
High work hardening at elevated temperature.

**Figure 10 materials-15-04348-f010:**
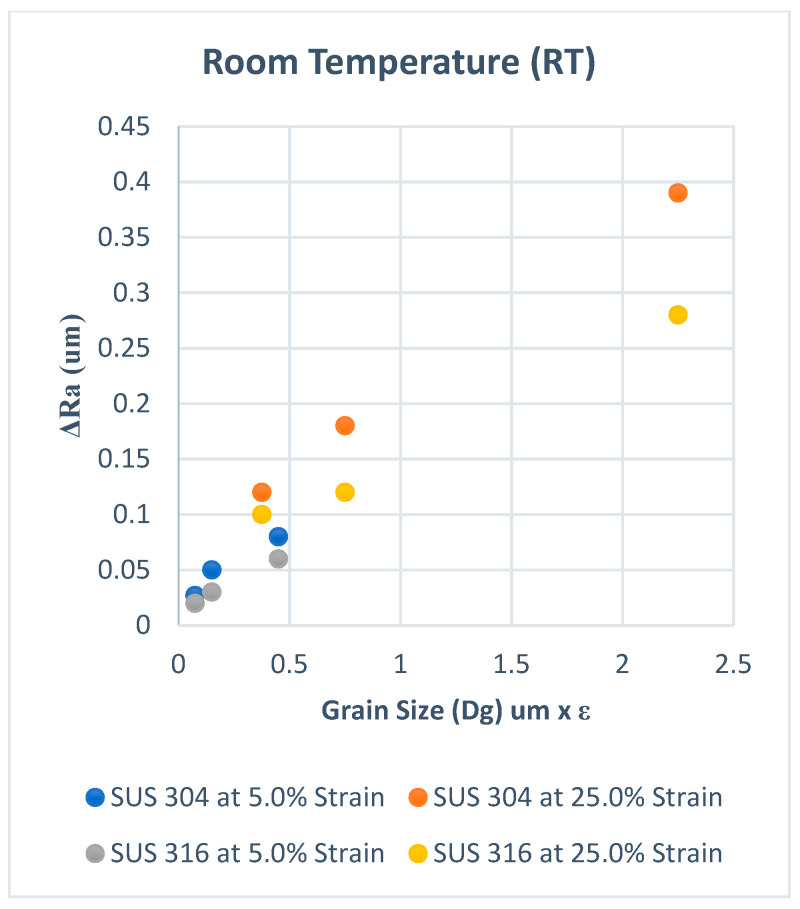
ΔRa versus Dg × ε at room temperature (RT).

**Figure 11 materials-15-04348-f011:**
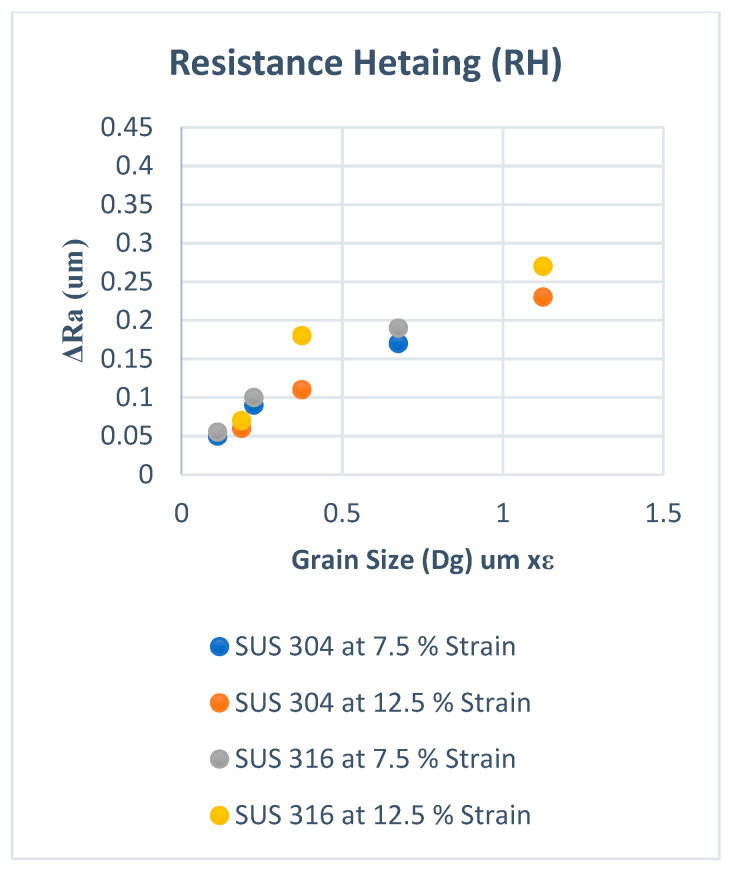
ΔRa versus Dg × ε at resistance heating (RH).

**Figure 12 materials-15-04348-f012:**
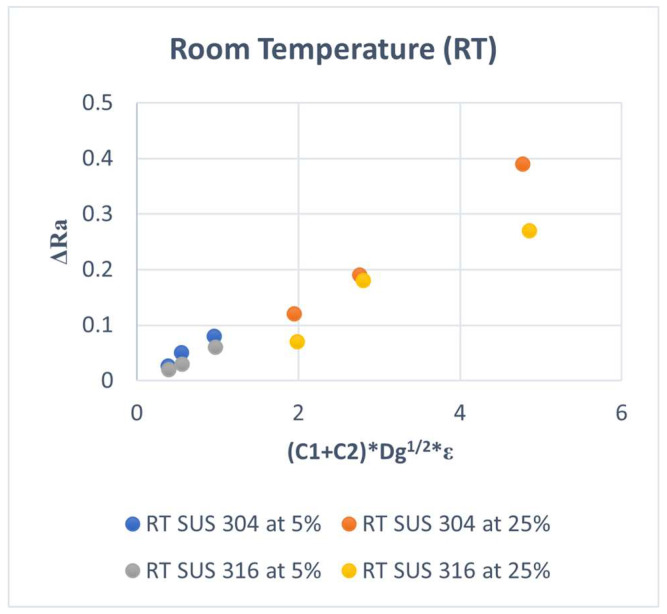
ΔRa versus (C1 + C2) × g^1/2^ × E at RT.

**Figure 13 materials-15-04348-f013:**
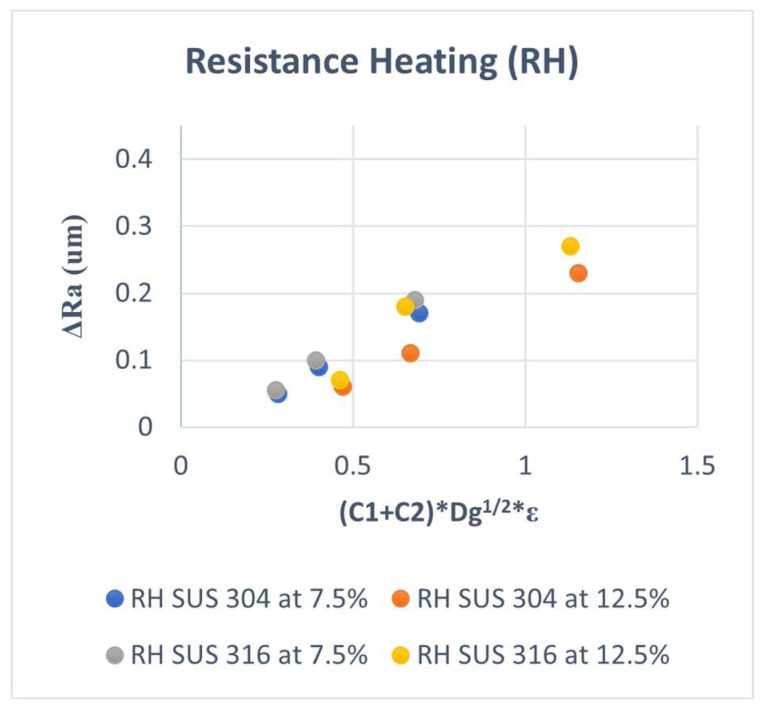
ΔRa versus (C1 + C2) × g^1/2^ × E at RH.

## Data Availability

Not applicable.

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
