# Peer review of "Constitutive Model of the Surface Roughening Behavior of Austenitic Stainless Steel"

_materials, 2022, doi:10.3390/ma15124348_

Round 1
Reviewer 1 Report
1.The currently recognized immediate factor cause of surface roughening is strain localization, which directly represented by density of Lower Angle Grain Boundaries (LAGBs), dislocation angles, Grain average misorientation (GAM) or Kernel average misorientation (KAM). It’s no any novelty that study of strain localization expressed by. grain misorientation (GMO).
2.The order of referenced literatures is out of sequence, and the format of references is not uniform.
3.We all know that the misorientation can be obtained by EBSD results, but in the method of this article, it is obtained using the im-age J technique utilizing the volume fraction based on color. Please confirm it.
4.Some expressions are inaccurate. For example, Figure 1 is a phase map without visible surface roughening. Values for coarse and fine grains are not given.
5.What does C in Equation 1 mean and what is the difference between ep. and ep?
Author Response
Dear Professor
Dear Sir
I would like to try to answer the question.
The sentence of "the surface roughening increase similar in fine grain of SUS 304 and SUS 316 TMF, because the MPT spread homogeneous (17), is omitted in line 6-7 in the subsection of 3.1. influence of volume fraction of MPT on surface roughening.
The answers are attached in this email.
Thanks Very Much for Professor
With Best Regard's
Aziz

Reviewer 2 Report
Very interesting article, but many changes should be made. The authors write about surface roughness measurements where they only refer to the previous article in the literature. More could be said about these measurements. Why was the focus only on the Ra parameter and why? This should be clarified. The depicted pictures of the structures should be larger as you can't see anything at the moment.
Author Response
Dear Professor
Dear Sir
I would like to try to answer the question from Professor.
Thanks Very Much for Professor.
With Best Regard's
Aziz

Round 2
Reviewer 2 Report
Thank you for the answers, but as we know the Ra parameter is the average surface roughness. How can this parameter affect the deformability of thin metal foils? Please prove it.
Author Response
We did not discuss affect of Ra to formability, but we discuss about micros train affect Ra. When thin metal foils subjected by plastic deformation, the Ra occur because of slip band movement. From the other researcher such as Furushima et al, Cheng et al, Fu et al, and so on concluded that Ra proportional to plastic strain. In this research, we want to use Ra is index to represent how the plastic deformation affect on surface roughening (Ra). Ra is kind of index and Ra is commonly use